# In Vitro Comparison of Sex-Specific Splicing Efficiencies of *fem* Pre-mRNA under Monoallelic and Heteroallelic Conditions of *csd*, a Master Sex-Determining Gene in the Honeybee

**DOI:** 10.3390/jdb11010010

**Published:** 2023-03-10

**Authors:** Yukihiro Suzuki, Takafumi Yamada, Masataka G. Suzuki

**Affiliations:** 1INTERSTELLAR Inc., 301 Unico A, 3-4 Nisshin-cho, Kawasaki-ku, Kawasaki 210-0024, Kanagawa, Japan; 2YAMADA-KUN’S Bee Farm, 95 Ochino, Mugegawa, Seki 501-2602, Gifu, Japan; 3Division of Biological Sciences, Department of Integrated Biosciences, Graduate School of Frontier Sciences, The University of Tokyo, 302 Bioscience-Bldg, 5-1-5 Kashiwanoha, Kashiwa 277-8562, Chiba, Japan

**Keywords:** *Apis mellifera*, complementary sex determiner, *feminizer*, sex determination, splicing

## Abstract

The sexual fate of honeybees is determined by the complementary sex determination (CSD) model: heterozygosity at a single locus (the CSD locus) determines femaleness, while hemizygosity or homozygosity at the CSD locus determines maleness. The *csd* gene encodes a splicing factor that regulates sex-specific splicing of the downstream target gene *feminizer* (*fem*), which is required for femaleness. The female mode of *fem* splicing occurs only when *csd* is present in the heteroallelic condition. To gain insights into how Csd proteins are only activated under the heterozygous allelic composition, we developed an in vitro assay system to evaluate the activity of Csd proteins. Consistent with the CSD model, the co-expression of two *csd* alleles, both of which lack splicing activity under the single-allele condition, restored the splicing activity that governs the female mode of *fem* splicing. RNA immunoprecipitation quantitative PCR analyses demonstrated that the CSD protein was specifically enriched in several exonic regions in the *fem* pre-mRNA, and enrichment in exons 3a and 5 was significantly greater under the heterozygous allelic composition than the single-allelic condition. However, in most cases *csd* expression under the monoallelic condition was capable of inducing the female mode of *fem* splicing contrary to the conventional CSD model. In contrast, repression of the male mode of *fem* splicing was predominant under heteroallelic conditions. These results were reproduced by real-time PCR of endogenous *fem* expression in female and male pupae. These findings strongly suggest that the heteroallelic composition of *csd* may be more important for the repression of the male splicing mode than for the induction of the female splicing mode of the *fem* gene.

## 1. Introduction

Animals that exploit sexual reproduction have evolved diverse sex determination mechanisms. For example, in most mammals sex determination is achieved using the XY-type mode, depending on the presence of the sex-determining region Y (Sry) gene on the Y chromosome [1]. The fruit fly *Drosophila melanogaster* also has X and Y chromosomes, but its sex is not determined by the presence of the Y chromosome [2]. Instead, the collective concentration of several X chromosome-encoded signal element (XES) proteins determines the active state of the *Sex-lethal* (*Sxl*) gene, which sits at the top of the sex determination cascade [3,4]. A sufficient amount of XSE is supplied only when the animal has two X chromosomes, leading to female sexual development. In contrast to mammals and *Drosophila*, the silkworm (*Bombyx mori*) has a sex chromosome composition in which females are ZW and males are ZZ, and individuals with the W chromosome differentiate into females [5]. The W chromosome-specific gene *feminizer* (*Fem*) produces Piwi-interacting RNA, which suppresses the expression of downstream male differentiation genes, leading to female development [6]. Thus, existing sex determination modes are diverse; however, in all of these species, the sex chromosome serves as a sex determination signal.

In contrast, hymenopteran insects do not have sex chromosomes that play a role in sex determination [7]. In these species, haploids hatched from unfertilized eggs become males, and diploids derived from fertilized eggs become females, but diploid males also occur at a low frequency. Whiting (1943) succeeded in creating a diploid male derived from a fertilized egg by inbreeding a *Habrobracon* species [8]. Based on the occurrence rate of diploid males, he proposed a model for sex determination called complementary sex determination (CSD). In this model, heterozygosity at a single locus (the CSD locus) determines femaleness in diploid individuals, while haploid individuals are hemizygous or homozygous for the *CSD* locus and thus develop into males [8]. The CSD model has been demonstrated in 33 hymenopteran species to date. Based on the frequency of diploid males, 9 *Csd* alleles have been estimated in *Habrobracon juglandis* [9], 19 in *Apis mellifera* [10], and approximately 50 in *Athalia rosae* [11].

The CSD locus was first molecularly identified in *Apis mellifera* [12]. Since then, 15 different *csd* alleles have been identified [13]. Sequence comparisons among *csd* alleles demonstrated that synonymous and non-synonymous substitutions occur at approximately the same rate, suggesting that this gene is neutrally evolving [14]. The knockdown of *csd* expression by RNA interference showed that a heterozygous allelic composition encodes a functional product that initiates female development, while a hemi- or homozygous allelic composition produces a nonfunctional product that leads to male development by default [12,15]. Beye et al. (2003) hypothesized that only a combination of polypeptides from different alleles yields an active heteromeric protein complex.

*csd* regulates sex-specific splicing of the downstream target gene *fem*, which is required for femaleness [15,16]. The female mode of *fem* splicing occurs only when *csd* is present in the heteroallelic condition. The female form of the Fem protein is active and induces female development [15,16]. In the absence of Csd protein activity in males, *fem* pre-mRNA is spliced into the male form, which includes male-specific exons (exons 3 to 5) with an intervening stop codon. The male form of Fem is inactive and leads to male development. As in other dipteran insects, the protein product of the *transformer-2* gene (*Am-tra2*) is essential for female-specific splicing of the *fem* pre-mRNA [17]. Nissen et al. postulated that the Am-Tra2 protein may act in combination with heteroallelic Csd proteins to mediate female *fem* splicing by binding to *fem* pre-mRNAs. However, the mechanism by which Csd proteins become active only when csd occurs in the heteroallelic composition remains unknown.

To identify this mechanism, we developed an in vitro assay system to evaluate the activity of Csd proteins. In the present study, the activity of Csd encoded by different alleles was quantitatively compared using that system. Our results demonstrate that Csd can induce the female mode of *fem* splicing even under single-allele conditions. Csd induced the female mode of *fem* splicing under heterozygous allelic conditions, but induced both the male and female modes of *fem* splicing in most cases under single-allele conditions. Based on these findings, we propose a new model for sex determination in honeybees.

## 2. Materials and Methods

### 2.1. Genomic DNA Extraction

Genomic DNA was extracted from adult honeybees, *Apis mellifera*, which were collected from the field at the University of Tokyo Kashiwa Campus (5-1-5 Kashiwanoha, Kashiwa, Chiba, Japan) around July 2016 according to a protocol reported previously [18]. 

### 2.2. RNA Extraction and RT-PCR

Total RNA was extracted from three adult females and from BmN cells using ISOGEN (Nippon Gene) reagent, as described previously [19]. Poly(A)+ RNA was further purified from the total RNA extracted from BmN cells using the Magnosphere UltraPure mRNA Purification Kit (TaKaRa), according to the manufacturer’s instructions. cDNA synthesis was performed according to a previously described protocol [19]. The resulting cDNA was subjected to PCR using KOD-FX (Toyobo) polymerase according to the manufacturer’s instructions. The primer sequences and PCR conditions utilized in this study are listed in Table 1. The PCR products were analyzed on a 2% agarose gel and visualized with ethidium bromide. 

### 2.3. Plasmid Construction

PCR amplification of the *fem* genomic sequence was accomplished using genomic DNA extracted from a single adult honeybee as a template. Initially, a 4973-bp fem gene fragment containing exons 2–6 was amplified by PCR using the primers 5′-CAT GGC GCG CCT AAC CTA AAA CAA GAT TTG CAG-3′ and 5′-CAT GTC GAC TGA TTT TTC AAT ATT TAC AGC-3′. The amplified DNA was cloned into the *Asc* I and *Sal* I sites immediately downstream of the *Bombyx mori cytoplasmic actin 3* promoter in pBmA3 [20] using the In-Fusion HD Cloning Kit (Takara Bio Inc.) according to the manufacturer’s protocol, thereby creating the *fem* minigene. The full-length open reading frame (ORF) of *csd* was amplified using the primers CsdBamF1 (5′-CGG GAT CCC ATG AAA CGA AAT ATA TCA AAT TAT TC-3′) and CsdHindR1 (5′-CCC AAG CTT CAT TGA TGC GTA GGT CCA AAT C-3′), with cDNA prepared from adult females as a template. The resulting product was digested at the *Bam*H I and *Hind* III sites, introduced using the CsdBamF1 and CsdHindR1 primers, and then inserted into the corresponding site of the pIEx1 vector (Novagen) to generate a plasmid for the expression of the Csd protein. The full-length ORF encoding the Am-Tra2 protein was obtained by PCR amplification using the primers Atra2NcoF (5′-CAT CCA TGG CAA TGA GTG ACA TTG AGC GAA G-3′) and Amtra2BamR (5′-CAT GGA TCC TTA ATA TCG ACG TGG TGA ATA AG-3′), with cDNA prepared from adult females as a template. The resulting product was digested at the *Nco* I and *Bam*H I sites, introduced using the Atra2NcoF and Amtra2BamR primers, and then inserted into the corresponding site of the pIEx1 vector to create a plasmid for the expression of Am-Tra2. The nucleotide sequences of the resulting constructs were confirmed by DNA sequencing. 

### 2.4. DNA Transfection of Cells 

The *fem* minigene (1 μg) was transfected into BmN cells established from the ovaries of the silkworm *B. mori* [19], with or without 1 μg plasmid DNA containing Csd protein-coding cDNA or Am-Tra2 protein-coding cDNA according to a protocol described previously [20]. Poly(A)+ RNA was isolated 48 h after transfection, as described above, and then subjected to cDNA synthesis according to a previously described protocol [20]. The female and male modes of *fem* splicing were detected by PCR using KOD-FX (Toyobo) polymerase according to the manufacturer’s instructions. The primer sequences and PCR conditions are listed in Table 1. All PCR products were analyzed on a 2% agarose gel and visualized with ethidium bromide. 

### 2.5. Quantitative RT-PCR 

Quantitative RT-PCR was performed according to a protocol published previously [20], using the primers listed in Table 2. Primers ArEF-1qF and ArEF-1qR, which specifically amplify the *elongation factor-1α* (*EF-1α*) gene in *Apis mellifera*, were used as the standard for quantification by PCR. The threshold cycle (CT) value was normalized to that of the *EF-1α* gene using Multiple RQ software (Takara Bio Inc., Kusatsu, Japan). We checked the dissociation curves of qPCR products to confirm that the primer sets did not produce off-target products. The expression of the target of interest (TOI) gene (female type *fem* [*fem-F*] and *csd*) was determined relative to a reference gene (*EF-1α*) over six qPCR runs. Finally, the level of the *fem-F* transcript relative to the *csd* gene was estimated in each group. The primer sequences are shown in Table 2.

### 2.6. RNA Immunoprecipitation qPCR (RIP-qPCR) Analysis

The full-length ORFs of the indicated *csd* alleles (*csd*^2-3^ and *csd*^3-1^) were artificially synthesized using by Eurofin Genomics (Tokyo, Japan). The synthesized DNA fragments were inserted into the *Nco* I and *Bam*H I sites in the pIEx-1 vector (Novagen) to create a plasmid for the expression of either CSD^2-3^ or CSD^3-1^ protein. The nucleotide sequence encoding V5-tag (GKPIPNPLLGLDST), which was linked to the 5’ end of the ORF of the indicated *csd* alleles, was also artificially synthesized as described above. The resulting DNA fragment was inserted into the *Nco* I and *Bam*H I sites in the pIEx-1 vector (Novagen) to create a plasmid for the expression of either V5-CSD^2-3^ or V5-CSD^3-1^ protein. The nucleotide sequences of the resulting constructs were confirmed using DNA sequencing. BmN cells were transfected with the *fem* minigene, together with plasmids that expressed CSD^2-3^, CSD^3-1^, V5-CSD^2-3^, and V5-CSD^3-1^ proteins in the indicated combinations. BmN cells transfected with the *fem* minigene and the empty vector (pIEx-1) served as negative controls. Three days after transfection, the cells were harvested and whole cell lysates were prepared according to a previously described protocol [21,22]. Then, 20 μL of Dynabeads Protein G (Invitrogen, Waltham, MA, USA) was incubated with 3.5 μg of anti-V5 tag antibody (ProteinTech Group, Rosemont, IL, USA) at room temperature for 10 min with gentle agitations. The resulting Dynabeads were incubated with cell lysate containing 400 μg of proteins at 4 °C for 1 h on a rotating wheel to prepare RIP samples. Negative control RIP samples were prepared using Dynabeads coupled with Normal Rabbit IgG (Fujifilm Corp., Tokyo, Japan). Pellets were washed five times with lysis buffer [23], and dissolved in 500 μL of ISOGEN during the last wash. RNA was isolated as described above, resuspended in 20 μL of RNase-free water, and treated with TURBO DNase (Thermo Fisher Scientific, Waltham, MA, USA) according to the manufacturer’s instructions. To quantify the amount of target sequence in the immunoprecipitated RNA/protein complexes, quantitative real-time PCR (qRT-PCR) was performed as described above. The primer sequences are shown in Table 2. 

## 3. Results

### 3.1. Acquisition of csd Alleles

The *csd* gene is a paralog of *fem*, which is an ortholog of the *transformer* (*tra*) gene [11]. *tra* is a conserved upstream component of the insect sex determination cascade that induces female development by regulating the sex-specific alternative splicing of downstream targets such as *doublesex* (*dsx*) [24,25]. As the silkworm does not have a *tra* ortholog, cultured cells established from silkworm tissues have been used to evaluate the function of tra in various animal species [20,21,26]. Therefore, in this study we aimed to assess the effect of Csd proteins on *fem* splicing using a previously established in vitro splicing assay system in cultured silkworm cells [20,21,26].

The *csd* alleles analyzed here were acquired from wild females (workers) of the honeybee, *Apis mellifera*, collected while flying around the University of Tokyo Kashiwa Campus in 2016. From those bees, we successfully obtained nine alleles, each of which encodes a different amino acid sequence. The alignment of the amino acid sequences of the Csd proteins encoded by each allele revealed mutations including amino acid substitutions and deletions at specific positions within the *N*-terminal region, the arginine- and serine-rich (RS) domain, the hypervariable region, and the proline-rich region (Figure 1a). To visualize the differences among these Csd proteins, mutations held by two representative alleles (*csd*^2-3^ and *csd*^2-6^) were defined as type A and type B, and the remaining alleles were classified by examining whether the mutations found at each amino acid position aligned with type A or type B (Figure 1b). For example, the Csd protein encoded by the *csd*^3-1^ allele consists of the same amino acid sequence as the Csd protein encoded by the *csd*^2-3^ allele, except for a mutation at amino acid position 336. For the strong expression of the Csd proteins in cultured silkworm cells, cDNA fragments containing ORFs derived from the respective *csd* alleles were incorporated into the pIEx-1 vector, which was designed to allow protein expression in cultured insect cells.

### 3.2. Heterogeneous Expression of Csd Proteins Induces Female-Specific Splicing of the fem Minigene

*fem* pre-mRNA exhibits sex-associated differences in splicing between exons 3 and 6. Female-specific *fem* splicing occurs between the splice donor site, located in the middle of exon 3, and the 5′ splice acceptor site of exon 6 (Figure 2a) [13]. To test whether our in vitro splicing assay system can reproduce the sex-specific splicing of *fem* pre-mRNA, the splicing pattern of the *fem* minigene was investigated by RT-PCR in the presence or absence of the Csd protein. For this purpose, the *fem* minigene, which contains genomic DNA encompassing exons 2 to 6 of *fem* (Figure 2b), was co-transfected with an expression vector encoding the Csd protein and the Am-Tra2 expression vector into BmN cells. The *csd* alleles used in these experiments were *csd^2-6^* and *csd^5-4^*. 

To avoid the detection of contaminating plasmid DNA in total RNA samples and to allow independent quantification of sex-specific splicing isoforms of the *fem* mRNAs, we designed primers that specifically anneal to the female-specific and male-specific *fem* transcripts (Figure 2b). RT-PCR analysis using these primers revealed that the female mode of *fem* splicing occurred when cells were transfected with both the Csd^2-6^ and Csd^5-4^ expression vectors (Figure 2c, lane 2). The same results were obtained when cDNA prepared from a female (=heterozygous for the *csd* allele) pupa was subjected to the same RT-PCR analysis (Figure 2c, lane 5). In addition to the female mode of *fem* splicing, the male splicing was observed when cells were transfected with a single Csd expression vector (Figure 2c, lanes 3 and 4). The same results were obtained by RT-PCR using cDNA prepared from a male (=hemizygous for the *csd* allele) pupa (Figure 2c, lane 6). On the other hand, no amplified products were observed in the negative control cells, into which the *fem* minigene was co-transfected with an empty vector instead of the Csd expression vector (Figure 2c, lane 1). These results demonstrated that our in vitro splicing assay system reproduced the endogenous sex-specific splicing regulation of *fem* pre-mRNA.

### 3.3. Csd Induces the Female Mode of Fem Splicing under Single-Allele Conditions

A previous study indicated that the female mode of *fem* splicing occurs only when *csd* is present in the heteroallelic condition [15]. Contrary to this finding, our RT-PCR analysis demonstrated that the female mode of *fem* splicing was observed even when the *fem* minigene was co-transfected with a single Csd expression vector (Figure 2c, lanes 3 and 4). The female mode of *fem* splicing was also observed in the male pupa that should be hemizygous for the *csd* allele (Figure 2c, lane 6). In our in vitro assay system, 1 μg of a single Csd expression vector was used for each assay. Thus, the amount of the Csd expression vector under the heteroallelic condition was twice as high as that under the single-allele condition, because two different Csd expression vectors were transfected into cells in the former case. This difference in the amount of transfected DNA may have led to a difference in the efficiency of *fem* splicing. To examine this possibility, we investigated the dose effect of Csd expression vectors on *fem* splicing. As shown in Figure 2d, transfection of 125, 250 and 500 ng Csd^2-6^ expression vector led to the female mode of *fem* splicing (Figure 2d, lanes 1 to 3), suggesting that at least 125 ng vector was required to induce the female mode of *fem* splicing. Similar results were obtained with the Csd^5-4^ expression vector (Figure 2d, lanes 4 to 6). Only the female *fem* splicing was observed when Csd^2-6^ and Csd^5-4^ expression vectors were transfected simultaneously, even though the amount of each vector was half that of that in the single allelic conditions (Figure 2d, lanes 7 to 9). These results suggest that no additive effects on the induction of female-mode *fem* splicing occurred among different *csd* alleles.

To assess whether Csd derived from a single allele can reproducibly induce female splicing, all *csd* alleles acquired in this study were subjected to in vitro analysis. As shown in Figure 3a, seven of the nine examined alleles induced the female mode of *fem* splicing. These results strongly suggest that Csd can induce the female mode of *fem* splicing, even under single-allele conditions. On the other hand, no amplification product was observed in cells transfected either with the Csd^2-3^ or Csd^3-1^ expression vectors (Figure 3a, upper panel, lane 1 and lane 3). These splicing defects could be attributed to the amino acid sequences of the protein products, as the sequences of these two alleles were identical aside from one amino acid residue at position 336 (described as H4 in Figure 1a). Interestingly, the splicing activity that leads to the female mode of *fem* splicing was fully restored when these two alleles were expressed simultaneously (Figure 3b, upper panel, lane 7). The female mode of *fem* splicing was also observed when cells were transfected with Csd expression vectors in all heteroallelic compositions examined (Figure 3b, upper panel). However, the male mode of *fem* splicing was observed even in several heteroallelic compositions (Figure 3b lower panel, allelic compositions 2-3/2-6, 3-2/3-7, 2-3/3-1, and 2-3/3-2).

These results suggest that heteroallelic conditions are not necessary to induce the female mode of *fem* splicing. However, it is possible that the efficiency of splicing may be greater under the heteroallelic than single-allele condition. To test this hypothesis, the level of female type transcript produced from the *fem* minigene was quantified and compared between these two conditions by qRT-PCR. The relative levels of female type *fem* transcript varied significantly among *csd* alleles (Figure 3c). In most cases, Csd expression in the heteroallelic composition yielded similar levels of the female-type transcript to the cells that expressed Csd^2-6^, Csd^3-2^, and Csd^3-7^ in the single-allele condition (Figure 3c). Only two heteroallelic compositions showed significantly higher levels of female-mode *fem* splicing than the single-allele condition (Figure 3c; allelic compositions Csd^2-3^/Csd^2-6^ and Csd^2-3^/Csd^3-7^). These results strongly suggest that the heteroallelic composition of *csd* is not necessary for improving the efficiency of female-mode *fem* splicing. However, the level of the female type *fem* transcript in heteroallelic compositions was significantly higher than that in monoallelic conditions, when all the data were integrated and compared between the two groups (Figure 3d).

### 3.4. Csd Protein Specifically Enriches fem Pre-mRNA Even under Monoallelic Conditions

The above results strongly suggest that the heteroallelic composition of *csd* is not necessary to induce the female mode of *fem* splicing, and that the efficiency of *fem* splicing is similar between monoallelic and heteroallelic conditions. To validate the results, we assessed the interaction between Csd proteins and *fem* pre-mRNA, and compared the efficiency of RNA-protein complex formation between monoallelic and heteroallelic conditions.

Nissen et al. (2012) postulated that the Am-Tra2 protein may act in combination with heteroallelic Csd proteins to mediate female fem splicing by binding to fem pre-mRNAs [17]. In *D. melanogaster*, Tra and Tra2 proteins induce female-specific splicing of *dsx* by binding to repetitive cis-elements, called dsxREs, that present on female-specific exons of *dsx* [27,28]. Comparative sequence analyses of *A. mellifera dsx* and *Nasonia vitripennis dsx* revealed the presence of (U/G)GAAGAU(U/A) repeats, clustered on a short sequence stretch in female-specific exons [29]. Bertossa et al. (2009) suggested that repeats of this consensus sequence may constitute recognition sites for the factors activating female-specific splicing in hymenopteran species [29]. We searched for the core sequence (GAAGAU) of the putative female-specific splicing factor-binding sites in the *fem* pre-mRNA sequence, and found that the *fem* pre-mRNA had five GAAGAU sites (described as e2, e3a, e3b, e5, and i5 in Figure 4a). A comparison of nucleotide sequences flanking these five sites with the 14-nt consensus sequence identified previously [29] revealed that 11-nt of the 14-nt consensus sequences were identical to e3a and e5 (Figure 4b). RIP-qPCR analysis was performed to examine whether Csd proteins interacted with the fem pre-mRNA through these five sites. *csd*^2-3^ and *csd*^3-1^ were subjected to the analysis because the female mode of *fem* splicing was observed only when these two alleles were expressed under heteroallelic conditions (Figure 3a,b). Since we had not prepared an antibody against Csd, we produced Csd fused to the V5 tag and immunoprecipitated Csd-RNA complexes with anti-V5 tag antibodies (see Materials and Methods). Then, the RNA was extracted and the enrichment of V5 tag-fusion protein on the *fem* pre-mRNA was quantified using qRT-PCR with primers that specifically amplified regions containing the five GAAGAU sites (Figure 4a). The exonic sequence of exon 3a was almost identical to that of exon 5, and we could not design primers that allowed discrimination between exons 3a and 5. The highest level of protein enrichment on exonic regions containing e3a and e5 sites was observed when cells were transfected with Csd^2-3^ expression vector together with Csd^3-1^ expression vector (heteroallelic composition) (Figure 4c). Although single allelic expression of Csd^2-3^ and Csd^3-1^ decreased the level of protein enrichment in the same region, enrichment was still significantly higher than in cells transfected with the empty vector. These results suggested that heteroallelic expression of the Csd protein may allow for interaction between Csd protein and *fem* pre-mRNA, leading to the female mode of *fem* splicing. However, contrary to the hypotheses of Beye et al. (2003) [12] and Nissen et al. (2012) [17], the formation of heteromeric Csd protein complexes seemed unnecessary for Csd-*fem* pre-mRNA interaction, because a sufficient level of Csd enrichment was observed in monoallelic conditions. Significant enrichment was also observed on the exonic region containing e2 site when cells were transfected either with Csd^2-3^ or Csd^3-1^ expression vector (Figure 4c). Heteroallelic expression of Csd alleles caused no significant enrichment in the same region. An interaction between the Csd protein and the *fem* pre-mRNA was observed regardless of whether the Csd expression was under monoallelic or heteroallelic conditions, consistent with the observation that the female mode of *fem* splicing occurred even under single allele conditions. In addition, the fact that Csd^2-3^ and Csd^3-1^ could interact with *fem* pre-mRNA despite a lack of *fem* splicing activity (Figure 2a,c) indicated that the interaction between Csd and *fem* pre-mRNA may not be directly involved in the female mode of *fem* splicing.

## 4. Discussion

To molecularly characterize the function of Csd, which is responsible for CSD in honeybees, we developed an in vitro assay system to evaluate the splicing activity of Csd proteins (Figure 2). In the present study, we used this system and found that *csd* expression in the monoallelic condition was capable of inducing the female mode of fem splicing (Figure 2 and Figure 3). The efficiency of female-mode *fem* splicing was similar between single-allele and heteroallelic conditions (Figure 3c). Interactions between Csd protein and *fem* pre-mRNA were observed regardless of whether the Csd expression was monoallelic or heteroallelic (Figure 3c). The most striking difference between monoallelic and heteroallelic Csd expression was the occurrence (or not) of the male mode of *fem* splicing (Figure 3a,b). The male mode of *fem* splicing was generally observed only under monoallelic conditions (Figure 3a). Consistent with our findings, RT-PCR analysis in a previous report indicated the presence of both *fem-F* and *fem-M* transcripts in haploid male embryos [17]. These findings are in contrast to a previous report that Csd proteins are activated only when *csd* occurs in the heteroallelic composition, leading to female development via the induction of female-mode *fem* splicing [13,15]. One possible explanation for this discrepancy is that the male form of the Fem protein may play a role in promoting male development. Alternatively, the male-specific isoform of Fem may act as a dominant negative regulator of the female Fem isoform, which is required for femaleness, thus leading to male development. *fem* has been identified as a *tra* ortholog in *A*. *mellifera* [13]. The splicing patterns of *tra* transcripts characterized to date in holometabolous insects indicate sexual dimorphism, with only the female-specific splicing variant encoding a functional protein, while the male-specific splicing variant yields truncated or nonfunctional protein products lacking significant domains [30] (Figure 5a). The male-mode *fem* transcript produces a truncated version of Fem in *A*. *mellifera* due to the inclusion of exons containing an intervening stop codon [13]. However, in contrast to the male Tra proteins in other holometabolous insects, the male isoform of Fem in *A*. *mellifera* contains an RS domain, which mediates protein–protein interactions and regulates the recognition of specific splice sites [31] (Figure 5a). Thus, the male isoform of Fem may have some function in male development in the honeybee. This hypothesis is supported by the previous finding that Am-Tra2 plays a positive role in inducing the male mode of *fem* splicing [17].

If the male isoform of Fem has functions in male development, then *csd* expression under heteroallelic conditions would be more important for repressing the male mode of splicing than for inducing the female mode of splicing. Beye et al. (2003) hypothesized that only the combination of polypeptides from different alleles yields an active heteromeric protein complex, as most of the allelic polymorphism in the Csd amino acid sequence exists within the C-terminal region, which contains the RS and proline-rich domains [12]. RS domains and proline-rich regions have protein binding capacity [32,33], suggesting that differences in the amino acid sequences of these regions may affect the protein−protein interactions involved in splicing regulation. Csd^2-3^ and Csd^3-1^ were capable of interacting with *fem* pre-mRNA under monoallelic conditions despite a lack of *fem* splicing activity (Figure 4c). This suggests that the functional difference between monoallelic and heteroallelic Csd proteins was not due to the ability to bind to the *fem* pre-mRNA, but rather to the ability to interact with factors required for regulating sex-specific *fem* splicing. Further research is needed to clarify whether monoallelic and heteroallelic Csd proteins show differences in protein profiles interacting with Csd.

In several heteroallelic compositions the male mode of *fem* splicing was observed, as was the case under single-allele conditions (Figure 3b, allelic compositions 2-3/2-6, 3-2/3-7, 2-3/3-1, and 2-3/3-2). If our hypothesis is true, then individuals carrying these alleles will develop into males or intersex animals despite possessing the *csd* gene in the heteroallelic condition. In such cases, the resulting males would be diploid. However, diploid males do not participate in reproduction, as they are usually eaten by adult female worker honeybees during the larval stage [35,36]. As a consequence, it is not possible to find diploid males that are heterozygous for csd in nature.

In the present study, we identified two *csd* alleles (*csd*^2-3^ and *csd*^3-1^) that lacked the capacity to splice fem pre-mRNA into the female form (Figure 3a). The alignment of the amino acid sequences of Csd proteins revealed that four amino acid residues were common to both the *csd*^2-3^ and *csd*^3-1^ alleles and differed from other alleles, all of which were located in the RS domain (Figure 1a,b). This difference may underlie the loss of splicing activity. However, the same amino acid substitutions were observed in the *csd*^3-7^ allele, which showed relatively strong activity for inducing female-mode *fem* splicing (Figure 1a and Figure 3c). One possible explanation for this discrepancy is that three amino acid residues near the *N*-terminal region (amino acid positions 69, 73, and 115) in the *csd*^3-7^ allele differ from the sequences of *csd*^2-3^ and *csd*^3-1^ (Figure 1a,b). The *N*-terminal region of Csd may include some unknown domains that are capable of complementing the function of the RS domain. The amino acid substitutions specifically observed in the *csd*^3-7^ allele may enhance this complementary activity, thereby restoring the loss of function of the RS domain.

As described above, *csd*^2-3^ and *csd*^3-1^ showed no splicing activity under the single-allele condition (Figure 3a,c). Nevertheless, co-expression of these two alleles fully restored the splicing activity that governs the female mode of *fem* splicing (Figure 3b,c). A single amino acid sequence polymorphism between these two alleles was found within the hypervariable region (H4 in Figure 1a), which has been identified previously [12]. Beye et al. (2003) reported that most amino acid differences characterizing the various alleles of this gene are found in the hypervariable region located between the RS domain and proline-rich region. However, whether these differences have functional significance remains unknown. Our results suggest that amino acid sequence polymorphisms in the hypervariable region may play an important role in restoring splicing activity when *csd* alleles with serious defects are present in a heterozygous individual. The heteroallelic expression of Csd^2-3^ and Csd^3-1^ slightly increased the relative Csd protein enrichment level on the *fem* pre-mRNA; this increase may help to restore the splicing activity. Although the details of the mechanism underlying this restoration are unknown at present, our findings provide important clues for understanding the function of the hypervariable region.

## 5. Conclusions

To gain insights into how Csd proteins are activated only under the heterozygous allelic composition, we developed an in vitro assay system to evaluate the activity of Csd proteins. By using this system, the splicing activity of Csd encoded by different alleles was quantitatively compared. Contrary to the conventional model as described above, most of the examined *csd* alleles were able to induce the female mode of *fem* splicing, even under monoallelic conditions. Moreover, our results clearly showed that a heterozygous allelic condition is critical for repressing the male mode of *fem* splicing, rather than for inducing female-specific *fem* splicing. These results were reproduced by RT-PCR analysis of endogenous *fem* expression in female and male pupae. These findings strongly suggest that the heteroallelic composition of *csd* may be more important for the repression of the male splicing mode than for the induction of the female splicing mode of the *fem* gene. In contrast to the male Tra proteins in other holometabolous insects, the male isoform of Fem in *A*. *mellifera* is relatively larger and contains a functionally important domain, RS domain (Figure 5). Thus, the male isoform of Fem may have some function in male development in the honeybee. Based on these findings, we propose a new model for sex determination in honeybees, where *csd* expression under heteroallelic conditions would be important for repressing the male mode of *fem* splicing to ensure the female development. Our hypothesis is supported by the previous finding that both females and males produce presumably functional Tra protein isoforms that include the functionally important domains in several hemimetabolous insects [37]. *fem* in the honeybee might still partially retain the ancestral state of the *tra* gene.

## Figures and Tables

**Figure 1 jdb-11-00010-f001:**
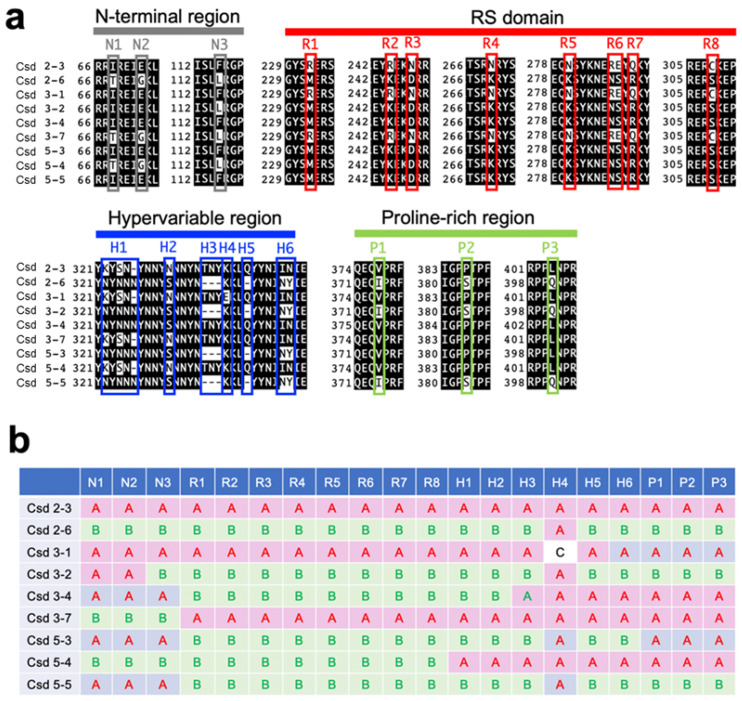
Variations in the amino acid sequences encoded by *csd* alleles identified in this study. (**a**) Alignment of the amino acid sequences in regions with interallelic polymorphisms among the alleles. Both the *N*-terminal region and proline-rich regions contain three amino acid substitutions (N1–N3 and P1–P3). Seven amino acid substitutions (R1–R5, R7, and R8) and a substitution of two consecutive amino acid residues (R6) were found within the RS domain. Various mutations were observed within the hypervariable region (R1–R8). (**b**) Classification of alleles based on the polymorphisms in their amino acid sequences. The mutations observed at the indicated positions (N1–N3, R1–R8, H1–H6, and P1–P3) in *csd^2-3^* and *csd^2-6^* were defined as types A and B, respectively. Polymorphisms that did not belong to either type A or B are classified as type C.

**Figure 2 jdb-11-00010-f002:**
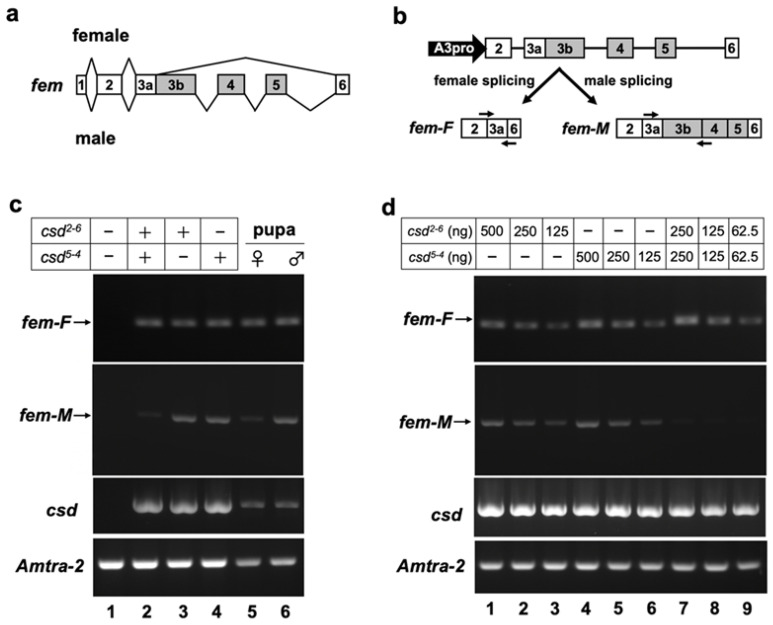
The in vitro assay system developed in this study for evaluating the splicing activity of Csd proteins. (**a**) Schematic diagram of sex-differential splicing of the *fem* gene reported by Hasselmann et al. (2008) [13]. Boxes represent exons. Introns are indicated by lines. The number in each box represents the exon number. Exons shown with gray boxes are male-specific exons. (**b**) Upper panel shows a schematic diagram of the *fem* minigene construct used for assessing the splicing activity of Csd protein in BmN cells. The filled arrow represents the promoter, consisting of a promoter sequence of the *Bombyx mori actin 3* gene (A3pro), used for inducing transcription of the *fem* minigene. Lower panel shows a female-specific *fem* transcript (*fem-F*) and a male-specific transcript (*fem-M*) produced by the two splicing modes. Arrows indicate the approximate position and direction of primers used for RT-PCR to detect the male and female splicing modes. (**c**) The *fem* minigene was co-transfected into BmN cells with a plasmid expressing the indicated *csd* allele. The Am-tra2 expression vector was also transfected in all cases. The female and male modes of *fem* splicing (f*em-F* and *fem-M*, respectively) were detected by RT-PCR using the primers illustrated in (**b**). The expression of *csd* and *Am-tra2* transfected into BmN cells was confirmed by RT-PCR. (**d**) The indicated amount of Csd expressing plasmid was transfected to examine the dose effect of the plasmid DNA on *fem* splicing. The RT-PCR conditions described in (**c**) were used to detect the female and male modes of *fem* splicing. The amplification products were separated on a 2% agarose gel and visualized using ethidium bromide.

**Figure 3 jdb-11-00010-f003:**
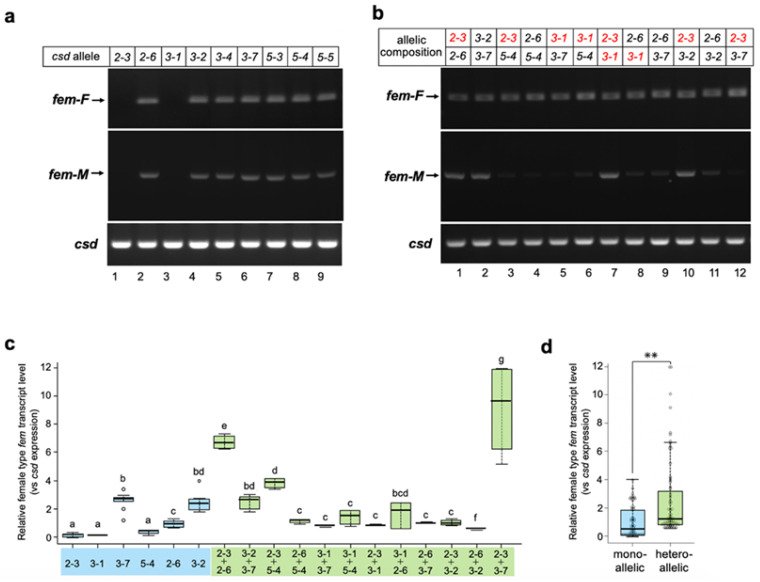
Splicing patterns of the *fem* minigene under single-allele and heteroallelic conditions. (**a**) All *csd* alleles acquired in this study were subjected to the analysis described in Figure 2. (**b**) Two different *csd* alleles in the indicated combinations were transfected into BmN cells with the *fem* minigene to recreate a heteroallelic condition. RT-PCR analysis was performed as described in Figure 2. The *csd* alleles lacking *fem* pre-mRNA splicing activity are shown in red. (**c**) The level of female-type *fem* transcript produced from the *fem* minigene relative to the *csd* expression level was compared between the monoallelic and heteroallelic conditions using qRT-PCR (Materials and Methods). The values obtained from each sample, and their distributions, are represented by box-and-whisker plots. Error bars are standard deviation. Different letters indicate statistically significant group differences (i.e., “a” is significantly different from “b” and “c”, etc.). “bd” means significantly different from groups other than b and d (i.e., not significantly different from groups b and d), and “bcd” means significantly different from groups other than b, c, and d (in other words, not significantly different from groups b, c, and d). There were six qPCR runs per experimental group; the Kruskal–Wallis test was followed by the Dunn–Bonferroni post hoc test, *p* < 0.001. (**d**) All single-allele and heteroallelic values indicated in (**c**) were integrated and statistically compared between the two groups. The values obtained for each sample, and their distributions, are represented by box-and-whisker plots. Error bars are standard deviation. ** *p* < 0.01 (Mann–Whitney U test).

**Figure 4 jdb-11-00010-f004:**
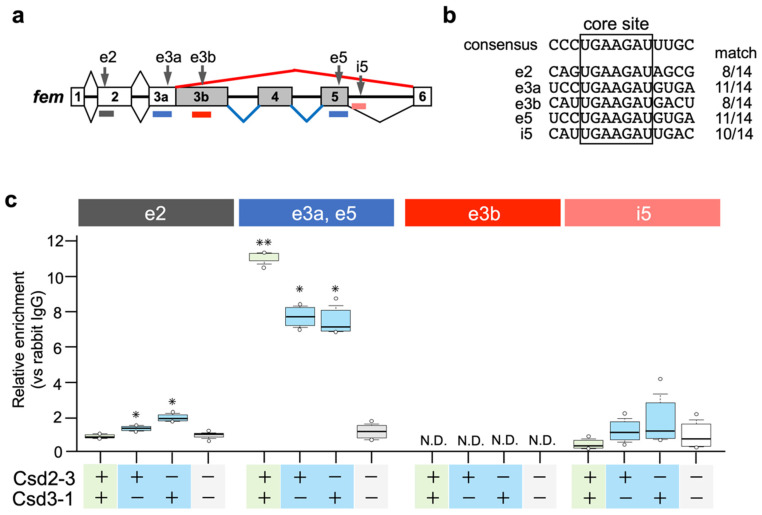
Csd protein is specifically enriched in the region containing putative binding sites for sex-specific splicing factors in the *fem* pre-mRNA. (**a**) Approximate positions of the putative binding sites for sex-specific splicing factors are indicated by vertical arrows (e2, e3a, e3b, e5, and i5). The distribution of qRT-PCR amplicons used in the RIP-qPCR analysis in (**c**) is denoted by colored horizontal lines. (**b**) Sequence comparison between each putative sex-specific splicing factor-binding site found in the *fem* pre-mRNA and the 14-nt consensus sequence described previously [29]. The number of matches is indicated on the right of each sequence. (**c**) The *fem* minigene was co-transfected into BmN cells together with Csd expression vectors (Csd^2-3^ and Csd^3-1^) in the indicated manner. RNA/V5 tag-fusion protein complexes were immunoprecipitated, and enrichment of the V5 tag-fusion protein on the *fem* pre-mRNA was quantified using qRT-PCR with primers that specifically amplified each region, as illustrated in (**a**). The relative enrichment was defined as the level of qPCR product amplified from samples relative to the samples precipitated using a negative control antibody (anti-rabbit IgG antibody). The values obtained from each group, and their distributions, are represented by box-and-whisker plots. There were six qPCR runs; * *p* < 0.05 and ** *p* < 0.01 compared to the negative control group (Csd^2-3-^ and Csd^3-1-^; cells transfected with empty vectors) in each region. One-way ANOVA was performed, with the Bonferroni post hoc test applied for multiple comparisons. N.D.: not detected.

**Figure 5 jdb-11-00010-f005:**
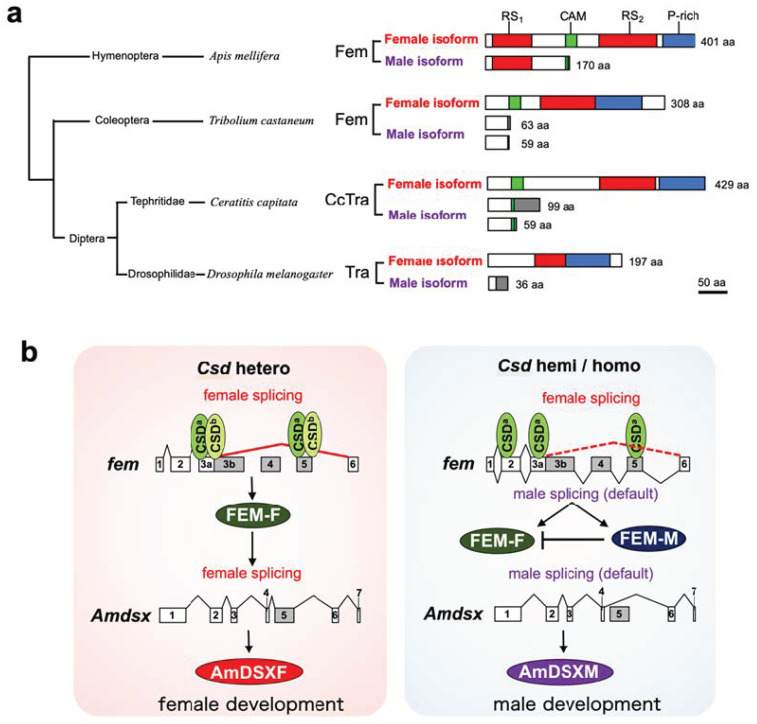
A unique feature of the male isoform of Fem in A. mellifera and its plausible function in males based on this study. (**a**) Schematic alignment of the male and female isoforms of Tra and its orthologous protein Fem. Arginine- and serine-rich domains (RS1 and RS2) are indicated in red; Tra-CAM domains (CAM), which are found in all Tra proteins except those of *Drosophila melanogaster* [34], are shown in green; and proline-rich domains (P-rich) are indicated with blue boxes. Gray boxes represent male-specific regions. Amino acid sequences of the female isoforms in the indicated organisms refer to ABU68668.1 in *Apis mellifera*, JQ857102 in *Tribolium castaneum*, XP_020714997.1 in *Ceratitis capitata*, and AAF49441.1 in *Drosophila melanogaster*. Amino acid sequences of the male isoforms refer to ABU68669.1 in *A*. *mellifera*, JQ857104 and JQ857103 in *T*. *castaneum*, AAM88675.1 and AAM88674.1 in *C*. *capitata*, and AAX53589.1 in *D*. *melanogaster*. (**b**) Schematic diagram of a novel CSD model proposed in this study. CSD proteins preferentially bind to the cis-regulatory elements on the exons 3a and 5 in the *fem* pre-mRNA under both heteroallelic and monoallelic conditions. The male-mode of the *fem* splicing is repressed only when *csd* is present in the heteroallelic condition, resulting in the production of FEM-F. FEM-F induces the female-specific splicing of the *A*. *mellifera dsx* (*Amdsx*) gene, yielding the female-isoform of AmDSX protein (AmDSXF). AmDSXF promotes female development. On the other hand, both male and female-modes of the *fem* splicing occur when individuals are hemizygous or homozygous for the CSD locus. The resulting FEM-M acts as a dominant negative regulator of FEM-F, thus leading to male development through the production of the male isoform of AmDSX protein (AmDSXM) that is produced by the default splicing of *Amdsx* pre-mRNA.

**Table 1 jdb-11-00010-t001:** Primer sequences and PCR conditions used for RT-PCR.

Gene	Primers	Sequence	Denaturation	Annealing	Elongation	N° Cycles
*fem-F*	femFF1	ACATTTATATTATCTGAAAAATTAG	98 °C	57 °C	68 °C	35
	femFR1	GCTTAGATCCTTCTCCCGTTC	10 s	30 s	30 s	
*fem-M*	femMF1	ATTAGAATCTTCAGATGGTAC	98 °C	57 °C	68 °C	35
	femMR1	TATGTAAAATTTAATATATTGCAC	10 s	30 s	30 s	
*csd*	csdF1	ATGAAACGAAATATATCAAATTATTC	98 °C	55 °C	68 °C	30
	csdR1	TCATTGATGCGTAGGTCCAAATC	10 s	30 s	90 s	
*Amtra-2*	Amtra2F	CAATGAGTGACATTGAGCGAAG	98 °C	55 °C	68 °C	30
	Amtra2R	TTAATATCGACGTGGTGAATAAG	10 s	30 s	60 s	

N° Cycles indicates number of cycles.

**Table 2 jdb-11-00010-t002:** Primer sequences used for RIP-qPCR.

Target	Primers	Sequence
e2	FemRIPF0	TTAGACAATCACGCAGTGAAGATAG
	FemRIPR0	CCCGTTCTTGTTGTATCATCCATTC
e3a and e5	FemRIPF1	GGACCAGAAGATACTCAAGTTAGTGC
	FemRIPR1	CCCGTTCTTCTTTTGAGCATCACAT
e3b	FemRIPF2	TCAGCAGAACTCGTCAAAATGTATGT
	FemRIPR2	GAAACAGGCCCGGAATGCAAAAG
i5	FemRIPF3	ACAGAACTCATCATTGAAGATTGAC
	FemRIPR3	TTAAATATAATTGGCGAGTTTTTGC
*EF-1α*	ArEF-1qF	TCTGTCGTGGCATCTTGAGC
	ArEF-1qF	TCTCCTGGGCTTCCTTCTCA
*fem-F*	femFqF1	AACATTTATATTATCTGAAAAATTAG
	femFqR1	GCTTAGATCCTTCTCCCGTTC
*csd*	csdqF1	ATGAAACGAAATATATCAAATTATTC
	csdqR1	TCATTGATGCGTAGGTCCAAATC

## Data Availability

The BmN cell lines used in this study are continuously passaged at the Laboratory of Bio-resource Regulation, Department of Integrated Biosciences, Graduate School of Frontier Sciences, The University of Tokyo. The male and female pupae of honeybees examined in this study is continuously reared at the YAMADA-KUN’S Bee Farm. All data obtained or analyzed during the present study are available from the corresponding author upon reasonable request.

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
