# Peer review of "In Vitro Comparison of Sex-Specific Splicing Efficiencies of fem Pre-mRNA under Monoallelic and Heteroallelic Conditions of csd, a Master Sex-Determining Gene in the Honeybee"

_jdb, 2023, doi:10.3390/jdb11010010_

Round 1

Reviewer 1 Report

Summary:

This is an excellently written, clear, and compelling article addressing the sex determination pathway in honeybees. This work will have an impact on the fields of evolution, development, and genetics. Specifically, the authors addressed the role of heterozygosity, hemizygosity, and homozygosity in the CSD locus in establishing sex specific isoforms of the gene feminizer. The authors performed a survey of the CSD alleles found in a wild population of honeybees for their ability to affect the splicing of feminizer in a silkworm cell culture system. Using RT-PCR they found that only some alleles were able induce both the male and female feminizer isoforms in a homoallelic condition, but all alleles could induce the female feminizer isoform in a heteroallelic condition. This indicated that the heteroallelic condition was necessary to induce the female feminizer isoform while suppressing the male feminizer isoform. It contrasted with previous work suggesting that the heteroallelic condition was necessary to induce the female feminizer isoform. They expanded on this performing RIP-qPCR to show that CSD was bound to putative female-specific splicing factor-binding sites in the feminizer pre-mRNA sequence in either the homoallelic or heteroallelic condition.

This work was comprehensive, clear, and convincing, adding to our understanding of sex determination systems. The figures were very clear and well labeled making them easy to digest. The experiments were well designed, and I was happy with the discussion of the potential caveats. I have a few minor concerns that would need to be fixed before publication, but these are only text and potentially a figure revision.

Minor Concerns:

Lines 39-40: the description of Drosophila melanogaster sex determination not up to date. Sex lethal is expressed based on X chromosome dosage (Erickson and Quintero 2007, Lu et. al. 2008) instead of the previous textbook description of X:A ratio. The model the authors referenced was updated, much like the impact the results of this manuscript will accomplish.

Erickson JW, Quintero JJ. Indirect effects of ploidy suggest X chromosome dose, not the X:A ratio, signals sex in Drosophila. PLoS Biol. 2007 Dec;5(12):e332. doi: 10.1371/journal.pbio.0050332. PMID: 18162044; PMCID: PMC2222971.

Lu H, Kozhina E, Mahadevaraju S, Yang D, Avila FW, Erickson JW. Maternal Groucho and bHLH repressors amplify the dose-sensitive X chromosome signal in Drosophila sex determination. Dev Biol. 2008 Nov 15;323(2):248-60. doi: 10.1016/j.ydbio.2008.08.012. Epub 2008 Aug 22. PMID: 18773886; PMCID: PMC2653429.

Lines 440-446: The figure 5 legend is a copy of the figure 4 legend and does not fit the figure.

Lines 91 and 192: adult honeybees are referenced in these lines. I appreciated the general name here for reference but would ask the authors to also add the species name for clarity.

Figure 5: I feel like this Figure could be switched for a more mechanistic graphic, highlighting that Csd heteroallelic conditions seem to reduce the male isoform of fem while the homoallelic condition allows both to the male and female isoform to be produced would be more helpful. It could leave the reader with an update of the current model of sex determination in honeybees and how the authors envision this result shaping role of the male specific isoform of fem in sex determination. However this is only a suggestion and is not necessary for publication of this article.

Author Response

Reviewer #1

Minor comments:

  1. Lines 39-40: the description of Drosophila melanogaster sex determination not up to date. Sex lethal is expressed based on X chromosome dosage (Erickson and Quintero 2007, Lu et. al. 2008) instead of the previous textbook description of X:A ratio. The model the authors referenced was updated, much like the impact the results of this manuscript will accomplish.

Erickson JW, Quintero JJ. Indirect effects of ploidy suggest X chromosome dose, not the X:A ratio, signals sex in Drosophila. PLoS Biol. 2007, 5:e332.

Lu H, Kozhina E, Mahadevaraju S, Yang D, Avila FW, Erickson JW. Maternal Groucho and bHLH repressors amplify the dose-sensitive X chromosome signal in Drosophila sex determination. Dev Biol. 2008, 323:248-260.

Answer: We thank for the reviewer's comments. According to the reviewer's instructions, we have added the following explanations.

The fruit fly Drosophila melanogaster also has X and Y chromosomes, but its sex is not determined by the presence of Y chromosome [2]. Instead, the collective concentration of several X chromosome-encoded signal element (XES) proteins determines the active state of the Sex-lethal (Sxl) gene, which sits at the top of the sex determination cascade [3, 4]. A sufficient amount of XSE is supplied only when the animal has two X chromosomes, leading to female sexual development.

  Please also see the lines 39 to 43 in the revised manuscript. We have also included the references listed by the reviewer in the above explanations as references "3" and "4".

  1. Lines 440-446: The figure 5 legend is a copy of the figure 4 legend and does not fit the figure.

Answer: We apologize for this careless mistake. We have replaced the relevant part with a legend that is appropriate for Figure 5. Please also see the lines 512-531.

  1. Lines 91 and 192: adult honeybees are referenced in these lines. I appreciated the general name here for reference but would ask the authors to also add the species name for clarity.

Answer: We thank for the reviewer's suggestions. We have added the species name for clarity to the corresponding parts. Please also see the lines 98 and 227.

  1. Lines 91 and 192: I feel like this Figure could be switched for a more mechanistic graphic, highlighting that Csd heteroallelic conditions seem to reduce the male isoform of fem while the homoallelic condition allows both to the male and female isoform to be produced would be more helpful. It could leave the reader with an update of the current model of sex determination in honeybees and how the authors envision this result shaping role of the male specific isoform of fem in sex determination. However this is only a suggestion and is not necessary for publication of this article.

Answer: We agree with the reviewer's comments. We have added a schematic diagram of a novel CSD model proposed in our study together with a legend of the figure. Pleas also see Figure 5b.

Reviewer 2 Report

In the manuscript "In vitro comparison of sex-specific splicing efficiencies of fem pre-mRNA under monoallelic and heteroallelic conditions of csd, a master sex-determining gene in the honeybee", Suzuki et al present a study in which they used a in vitro system to evaluate the splicing activity of monoallelic and heteroallelic Csd proteins. Interestingly, they found that Csd is able to induce female-mode splicing of the fem gene in both monoallelic and heteroallelic conditions, while the male-mode splicing is generally only observed in monoallelic conditions. They also reported interactions between Csd protein and fem pre-mRNA in both monoallelic or heteroallelic conditions, suggesting that the functional difference between the monoallelic and heteroallelic Csd proteins is not due to their ability to interact with the pre-mRNA. They propose that the heteroallelic Csd may be important for repressing the male-mode splicing of the fem gene, leading to female development.

Overall, the rationale for their study is well defined and their conclusion is strongly supported. I have a couple concerns as outlined below:

1. In Figure 1b, some of the background for type A are pink, and some are blue. What is the difference between the pink-background A and blue-background A?

2. When describing the dose effect of Figure 2d, the authors describe that “transfection of 250 and 500 ng Csd2-6 expression vector led to the female mode of fem splicing (Figure 2d, lanes 1 and 2)”, but in the figure there is also a band in the 125ng condition (lane 3). Similarly, they describe “In the case of the Csd5-4 expression vector, at least 500 ng vector was required to induce the female mode of fem splicing (Figure 2d, lane 4)”, but there are also bands in the 250 and 125ng conditions (lanes 5 and 6). Could the authors be more quantitative and accurate when describing the results in Figure 2d?

3. In line 464, it should be referring to Figure 3b, not 3c. The authors talk about the male-mode splicing results here. Could the authors also briefly talk about it in the Results part, or let readers know that there will be discussion about it later in Discussion part?

4. In figure 3c, the authors use different letters to indicate statistically significant group differences (i.e., “a” is significantly different from “b” and “c”, etc.). The “bd” and “bcd” groups are confusing. Is “b” significantly different from “d”? And is “bd”/”bcd” significantly different from “b” or “c” or “d”?

Minor points:

1. In line 203, the Csd3-1 is the same as Csd2-3 (type A), not Csd2-6.

2. In line 267, it is referring to Figure 2c lanes 2 and 3, while it should be lanes 3 and 4.

Author Response

Reviewer #2

Major comments:

  1. In Figure 1b, some of the background for type A are pink, and some are blue. What is the difference between the pink-background A and blue-background A?

Answer: We apologize for this careless mistake. The correct background color for type A is pink, so we changed all the background color for type A to pink. Please also see the Figure 1b in the revised manuscript.

  1. When describing the dose effect of Figure 2d, the authors describe that “transfection of 250 and 500 ng Csd2-6 expression vector led to the female mode of fem splicing (Figure 2d, lanes 1 and 2)”, but in the figure there is also a band in the 125ng condition (lane 3). Similarly, they describe “In the case of the Csd5-4 expression vector, at least 500 ng vector was required to induce the female mode of fem splicing (Figure 2d, lane 4)”, but there are also bands in the 250 and 125ng conditions (lanes 5 and 6). Could the authors be more quantitative and accurate when describing the results in Figure 2d? The bottom line is that these are key controls, without which we cannot have confidence in the conclusions of the study.

Answer: We thank for the reviewer's comments. According to the comments, we have modified the text appropriately according to the amount of each expression vector shown in Figure 2d. Please also see lines 321-329 in the revised manuscript.

  1. In line 464, it should be referring to Figure 3b, not 3c. The authors talk about the male-mode splicing results here. Could the authors also briefly talk about it in the Results part, or let readers know that there will be discussion about it later in Discussion part?

Answer: We thank for the reviewer's comments. We have changed the corresponding part from Figure 3c to Figure 3b. According to the reviewer's suggestion, we mentioned that allelic compositions 2-3/2-6, 3-2/3-7, 2-3/3-1, and 2-3/3-2 also induced the male-mode of fem splicing in the appropriate section of the results chapter. Please also see lines 343-345, and 589 in the revised manuscript.

  1. In figure 3c, the authors use different letters to indicate statistically significant group differences (i.e., “a” is significantly different from “b” and “c”, etc.). The “bd” and “bcd” groups are confusing. Is “b” significantly different from “d”? And is “bd”/”bcd” significantly different from “b” or “c” or “d”?

Answer: We apologize for the lack of explanation. “bd" means significantly different from groups other than b and d (i.e., not significantly different from groups b and d), and “bcd“ means significantly different from groups other than b, c, and d (in other words, not significantly different from groups b, c, and d). We have added these explanations to the legend of Figure 3c in the revised manuscript.

Minor comments:

  1. In line 203, the Csd3-1 is the same as Csd2-3 (type A), not Csd2-6.

Answer: We have corrected the points according to the reviewer’s comment. Please also see the line 237 in the revised manuscript.

  1. In line 267, it is referring to Figure 2c lanes 2 and 3, while it should be lanes 3 and 4.

Answer: We have corrected the points according to the reviewer’s comment. Please also see the lines 313-314 in the revised manuscript.